physiology

body size, heat tolerance, thermal tolerance, allometry, reversible plasticity

**Author for correspondence:**
Tim Burton
e-mail: tim.burton@ntnu.no

# Acclimation capacity and rate change through life in the zooplankton *Daphnia*

Tim Burton[1], Hanna-Kaisa Lakka[1,2] and Sigurd Einum[1]

[1]Centre for Biodiversity Dynamics, Department of Biology, Norwegian University of Science and Technology, Realfagbygget, NO-7491 Trondheim, Norway
[2]Department of Biological and Environmental Science, University of Jyväskylä, Jyväskylä, Finland

TB, 0000-0002-0215-0227; SE, 0000-0002-3788-7800

When a change in the environment occurs, organisms can maintain an optimal phenotypic state via plastic, reversible changes to their phenotypes. These adjustments, when occurring within a generation, are described as the process of acclimation. While acclimation has been studied for more than half a century, global environmental change has stimulated renewed interest in quantifying variation in the rate and capacity with which this process occurs, particularly among ectothermic organisms. Yet, despite the likely ecological importance of acclimation capacity and rate, how these traits change throughout life among members of the same species is largely unstudied. Here we investigate these relationships by measuring the acute heat tolerance of the clonally reproducing zooplankter *Daphnia magna* of different size/age and acclimation status. The heat tolerance of individuals completely acclimated to relatively warm (28°C) or cool (17°C) temperatures diverged during development, indicating that older, larger individuals had a greater capacity to increase heat tolerance. However, when cool acclimated individuals were briefly exposed to the warm temperature (i.e. were 'heat-hardened'), it was younger, smaller animals with less capacity to acclimate that were able to do so more rapidly because they obtained or came closer to obtaining complete acclimation of heat tolerance. Our results illustrate that within a species, individuals can differ substantially in how rapidly and by how much they can respond to environmental change. We urge greater investigation of the intraspecific relationship between acclimation and development along with further consideration of the factors that might contribute to these enigmatic patterns of phenotypic variation.

## 1. Introduction

Organisms can remodel their phenotypes within their own lifetimes to counteract potentially negative fitness effects of environmental change. This form of phenotypic plasticity, which is often reversible in nature, describes the process of acclimation or more generally, reversible plasticity [1,2]. Empirically, there has been substantial focus on quantifying and explaining variation in the capacity for such reversible phenotypic change because this provides an estimate of the degree to which organisms should be able to adjust to novel environments [3–8]. However, this focus on 'capacity' has meant that a key aspect of acclimation has been largely neglected, namely that it takes time. Thus, despite the presumed importance of acclimation in allowing organisms to persist in the face of environmental change, measurements of variation in the rate at which acclimation progresses within-species are surprisingly rare (some descriptive examples include [9,10–13]). This is somewhat puzzling, as estimates of acclimation capacity might take on greater ecological meaning if they can be considered alongside measurements of the rate at which acclimation proceeds. Here, we jointly consider how the capacity for and rate of acclimation change throughout life among members of the same species.

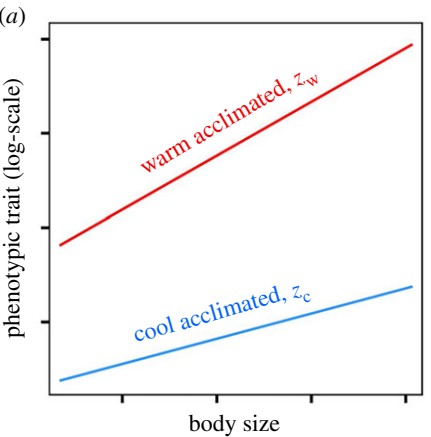
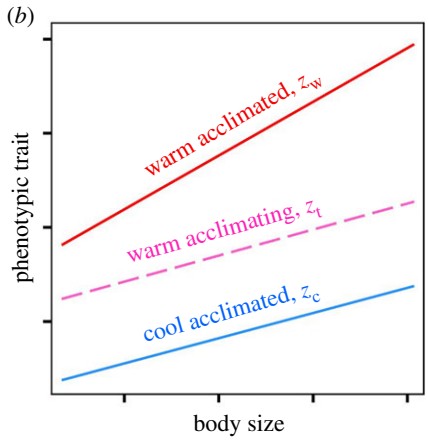

**Figure 1.** Conceptual figure illustrating definitions for acclimation capacity (*a*) and rate (*b*) adopted in the current study. Definitions are shown for a hypothetical phenotypic trait that is related positively to body size and has a positive acclimation response to temperature. (*a*) Acclimation capacity is the difference in trait value (on logarithmic scale to obtain a proportional difference) among individuals completely acclimated to warm ($z_w$, red line) and cool ($z_c$, blue line) temperatures. (*b*) Acclimation rate (to an increase in temperature) is the rate at which the trait changes from $z_c$ to $z_w$. This can be estimated by exposing cool acclimated individuals to the warm temperature for a given duration *t* that is insufficient to yield complete warm acclimation, then measuring the resulting phenotype $z_t$ of the partially acclimated individuals. The change in phenotype following partial acclimation is then expressed as a proportion of what is achievable under complete acclimation. Thus, acclimation rate is calculated as $(z_t - z_c)/(z_w - z_c) \times t$. Note that this calculation needs to be performed on the arithmetic scale to obtain proportionality. For the trait considered here, small individuals have a lower capacity for acclimation (i.e. smaller proportional difference in trait value when completely acclimated to the warm and cool temperatures) but can acclimate more rapidly (i.e. achieve a larger proportion of their acclimation capacity when exposed to the warmer temperature for a set period of time of a duration insufficient to obtain complete acclimation). (Online version in colour.)

The capacity for acclimation has generally been defined as the proportional difference in a given trait between individuals completely acclimated to different temperatures (*sensu*, [14] figure 1). However, the rate of acclimation may be considered in several ways. One option is to measure the absolute change in a trait per unit time (as adopted in comparative syntheses, e.g. [15]). However, the latter definition does not consider the amount of phenotypic change as a proportion of that possible in the completely acclimated state (figure 1). Crucially, data describing complete acclimation, which may often be unavailable in comparative syntheses, are simple to obtain via experimental manipulation. Assuming that completely acclimated individuals express phenotypes that yield the highest possible fitness for their genotype, then the rate at which they approach complete acclimation should give a measure of the expected duration during which the expressed phenotype is sub-optimal. Thus, when the rate of acclimation can be considered in this way, it arguably holds greater ecological relevance than the measures of acclimation rate made on an absolute scale.

In general, the relationship between development and acclimation is poorly understood. This gap in knowledge may be due in part to the difficulties inherent in measuring the relationship between acclimation and size/age within a species: acclimation takes time and over time individuals typically grow (especially early life stages), meaning that body size may also change during the treatment used to stimulate acclimation (most commonly a manipulation of temperature). However, this problem can be circumvented by studying traits that respond to environmental change with sufficient speed, such that an acclimation response can be measured before body size changes substantially. In this context, traits describing acute temperature tolerance may be promising because they often express a rapid response to temperature change, being detectable in multi-cellular organisms within as little as a few hours [16–18]. We used the widespread freshwater zooplankton species *Daphnia magna* to measure the acute heat tolerance of individuals of different body size and age that had experienced different temperature acclimation regimes. Based on these measurements, we quantified both the capacity for and the rate of acclimation. *Daphnia* have several characteristics that make them well suited for investigating the relationship between acclimation and development. First, they develop directly after hatching, meaning that differences in size/age are not confounded with distinct transitions in morphology or life stage (except for sexual maturation). Second, its facultatively clonal mode of reproduction precludes any measurement noise that might be caused by genetic variation among individuals. And finally, *Daphnia* can adjust heat tolerance rapidly, seemingly by 'tracking' variation in ambient temperature at a relatively fine temporal resolution [19]. This makes it possible to measure a relatively large acclimation response to temperature change in experimental subjects that have undergone little or no measurable change in size themselves.

To investigate the relationship between size/age and acclimation in *D. magna*, we measured the size scaling of heat tolerance among clonal replicates of a single genotype exposed to four different temperature treatments. In two of these treatments, measurements of heat tolerance were made on individuals that were completely acclimated to 17 or 28°C. These groups were used to estimate developmental effects on acclimation capacity by comparing the scaling relationship of heat tolerance among individuals completely acclimated to each temperature (figure 1*a*). In the two other groups, individuals initially acclimated to these same temperatures were exposed or 'hardened' (term describing acclimation response to a transient change in temperature that increases subsequent tolerance of more severe temperature exposures, [1]) to the opposite temperature for a brief period of fixed duration before the measurement of heat tolerance. The duration of hardening (approx. 10 h) was chosen to be *ca* 40% of the time period shown to best explain the variation in heat tolerance among adult *D. magna* that had developed in a series of treatments where acclimation status was likely to be in continual

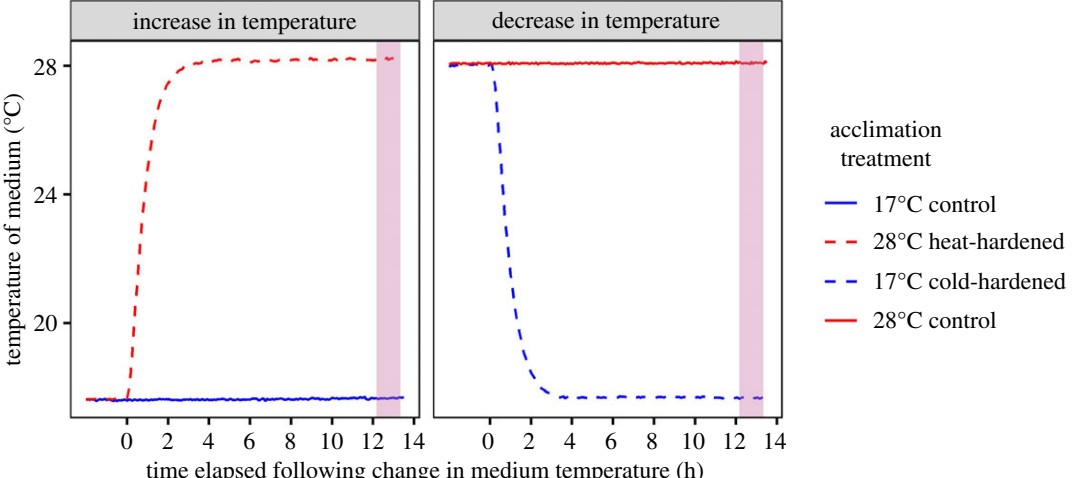

**Figure 2.** Time course of temperature change experienced by *Daphnia magna* in the heat- and cold-hardening treatments. Data are presented according to the temperature that individuals in the heat- and cold-hardening treatments were acclimated to initially. Thus, temperatures experienced by 28°C heat-hardened individuals are shown alongside those experienced by 17°C control individuals (left panel) and likewise for individuals from the 17°C cold-hardened and 28°C control treatments (right panel). The shaded region in each panel shows the variation among measuring runs in the time elapsed from the onset of temperature change in the hardening treatments until measurement of heat tolerance (range 12.2–13.3 h). (Online version in colour.)

flux [19]. In these treatments, temperature fluctuated in different temporal patterns (both among days and within them) and mean temperature experienced in the 24 h preceding measurement was found to be the best predictor of among-treatment variation in heat tolerance [19]. Thus, the chosen time period would ensure that an acclimation response would be measurable, but incomplete, a requirement for estimating the rate of acclimation. Developmental effects on acclimation rate (as depicted in figure 1*b*) were based on the heat tolerance of these 'hardened' individuals compared to that displayed by completely acclimated individuals.

## 2. Material and methods

### (a) Experimental cultures

Replicated clonal cultures of a single genotype of *D. magna* were maintained in 1.0 l glass beakers at either 17 or 28°C (hereafter = development temperature, 8 × cultures per temperature) in programmable climate cabinets (Memmert IPP, Germany). This genotype had been hatched previously from a resting egg collected in a shallow pond at Værøy Island, northern Norway (67.687°N, 12.672°E). The animals in these cultures were fed *ad libitum* amounts of a commercial shellfish diet (1800, Reed Mariculture Inc., USA) three times per week, in amounts that were specific to each development temperature. Medium (ADaM, [20]) in these cultures was changed once (17°C development temperature) or twice per week (28°C development temperature) when the number of individuals in each culture was culled to a haphazardly chosen selection of 15–20 animals. A 24 h day length mimicked the natural summer photoperiod of the source population (the pond from which the resting egg was collected is located more than 100 km north of the Arctic circle). The experimental genotype was acclimated to the developmental temperatures under these conditions for a minimum of four asexual generations before the onset of the temperature acclimation treatments. Each generation was initiated with juveniles born in second or later clutches in different beakers.

### (b) Acclimation to temperature change

After rearing clonal replicates of the experimental genotype under contrasting developmental temperatures, we subjected them to temperature manipulations (hereafter 'hardening treatments') that were designed to stimulate the physiological re-modelling that occurs in this species as it acclimates to temperature change. Individuals that varied in size were obtained by haphazardly selecting pairs of individuals from each of the eight replicate cultures that had been kept at each development temperature (smallest individuals chosen were approximately 24–48 h old). Within a pair, each individual was allocated to a separate 50 ml centrifuge tube with 45 ml of ADaM (same temperature as the development temperature) and provisioned with an *ad libitum* ration of shellfish diet that was specific to the temperature that each individual would experience by the conclusion of the forthcoming hardening treatments. For animals that developed at 17°C, one individual from each pair was subject to a control treatment (17°C control treatment) where the temperature remained constant at this value overnight. The other individual from the same pair was subject to an overnight period of heat-hardening (28°C heat-hardened treatment) where the medium temperature was increased gradually, plateauing at 28°C approximately 9–10 h before measurement (figure 2). The reverse of this procedure was repeated for pairs of animals that developed at 28°C to produce 17°C cold-hardened individuals and their respective controls. The same period was chosen for both hardening treatments so that size/age effects on any acclimation response would be more comparable. The temperatures implemented during development and the range in temperature experienced during hardening was within the range observed in the wild for the population from which the experimental genotype was sourced [21].

The following morning body sizes of the experimental animals were measured from digital images (using ImageJ, National Institutes of Health, Bethesda, MD) taken under a stereomicroscope as the length of the gut (anterior extremity of mid-gut to posterior extremity of the hindgut, image taken with the animal lying in a relaxed position, range 0.68–3.39 mm). During this period, when the experimental animals were removed from the climate cabinets, we were careful to maintain them at a temperature which corresponded to the value they had experienced at the end of the treatment period in their respective acclimation treatments. Following imaging, animals were then returned to their respective climate cabinets until measurement of heat tolerance. Published data describing the relationship between somatic growth and rearing temperature for the experimental genotype (and other genotypes from the same population, [22]), indicated that the

likelihood of obtaining a repeatable measurement of body size change over the brief hardening period employed here was low.

## (c) Heat tolerance

We measured heat tolerance as the ability to maintain bodily function at high temperature, recorded as the 'time to immobilisation' (referred to hereafter as $T_{imm}$) at 37°C [23,24]. $T_{imm}$ is a thermal endpoint defined by the time taken for locomotory function to cease at a constant, lethal temperature [25]. $T_{imm}$ was estimated using a modified version of an algorithm in the R computing environment that can objectively identify the loss of locomotory function from video-derived tracking data [23]. Individual daphnids from each of the acclimation treatments were exposed to 37°C using a custom-built, aluminium and glass thermostatic well plate (figure 1, [23]). In each measuring run, 30 individuals (7–8 individuals from each of the four acclimation treatments per run) were pipetted into the glass wells (arranged in a 5 × 6 configuration in the plate, well diameter 1.6 cm; well volume 4 ml, one individual per well) containing ADaM that had been pre-heated to 37°C. For each individual, we recorded the well number and time (in seconds) elapsed from the moment that the first individual was placed in a well (it took between 4 and 6 min to introduce all 30 individuals to the well plate). After the last individual had been pipetted into the well plate, it was filmed from above with a digital camera (Basler aCA1300-60gm, fitted with 5–50 mm, F1.4, CS mount lenses). Backlighting from an LED light board (Huion A4 LED light pad, set to maximum intensity) provided contrast between the individual in each well and the background. Video recording ceased when visual inspection indicated that all individuals were motionless. The resulting video files were processed in Ethovision (version XT 11.5, Noldus Information Technology, The Netherlands, settings: greyscale pixel range 10–145, pixel size range 4–350, sample rate 3 observations $s^{-1}$), to produce a time series of velocity data (in mm $s^{-1}$, travelled by the centre-point of each individual). Using this modified algorithm, we calculated the time taken (in minutes) for an individual's swimming velocity to reach a specified threshold value. In our experience, tracking software can still assess completely stationary individuals or objects as moving to some minor extent. We reasoned that the time until loss of normal mobility (i.e. our measurement of $T_{imm}$) should be evaluated relative to this baseline level of 'noise'. Thus, we recorded the maximum swimming velocity of each individual during the final 5 min of filming (where careful visual census had indicated complete cessation of swimming activity). These noise values represent a state of complete immobility as viewed by the tracking software (maximum = 0.4 mm $s^{-1}$). When calculating $T_{imm}$ for the analyses presented herein, we set the threshold swimming velocity in this algorithm as twice this maximum noise value (i.e. 0.8 mm $s^{-1}$). Altering this threshold value (by up to ±50%) had a negligible impact on the resulting parameter estimates and explanatory power of the model that was subsequently found to best fit the data (electronic supplementary material, figure S1). The modified algorithm for calculating $T_{imm}$ and tracking data are available from the Dryad Digital Repository: https://doi.org/10.5061/dryad.95x69p8g8 [26]. Up to two measuring runs (out of a total of 23) were performed on a given day between 10.30 and 14.30. There was minor variation among measuring runs in the time elapsed from the onset of temperature change in the two hardening treatments until the subsequent measurement of heat tolerance (hereafter 'treatment duration', range: 12.2–13.3 h, figure 2). In total, we obtained 678 individual measurements of $T_{imm}$: 173 individuals from the 17°C control treatment, 162 individuals from the 28°C heat-hardened treatment, 176 individuals from the 28°C control treatment and 167 individuals from the 17°C cold-hardened treatment. $T_{imm}$ could not be estimated for 12 individuals due to difficulty in acquiring accurate tracking data.

## (d) Data analysis

We employed linear mixed effect modelling to evaluate the relationship between $T_{imm}$ and body size (both variables transformed by taking natural logarithms) among individuals that were in the process of acclimating to a change in temperature relative to individuals who had developed under constant temperature (i.e. 28°C heat-hardened versus 17°C control individuals and 17°C cold-hardened versus 28°C control individuals). $T_{imm}$ and body size were transformed to natural logarithm scale so that (i) body size effects on acclimation capacity were modelled on a proportional scale (see [27,28] for discussion on utility of log-transformation in analyses of scaling phenomena) and (ii) assumptions of linear modelling were satisfied. We evaluated the relative support of four models (using AICc-based Akaike weights $w_i$, models fit with maximum likelihood), two of which considered the influence of body size and hardening treatment on heat tolerance as either interactive or additive, and two testing these same relationships but with treatment duration considered as an additional covariate. Given that individuals from each of the treatment and control groups were present in all 23 measuring runs, data from these four groups were directly comparable and analysed in a single model. To evaluate the relationship between body size and acclimation capacity, we relevelled the intercept term in the best-fitting model, comparing the 28°C control group relative to the 17°C control group.

Body size effects on the rate of acclimation were estimated using the best-fitting model to predict $T_{imm}$ values for each of the hardened and control groups over the range of body sizes for which a hardening response was observed (1.1–3.3 mm). Across this size range, predicted $T_{imm}$ values (back-transformed to arithmetic scale) were then used to calculate the acclimation shown by hardened individuals as a proportion of the total amount of acclimation possible (figure 1b). For instance, in response to an increase in temperature, the rate of acclimation (i.e. proportion of complete acclimation per approximately 10 h) of heat tolerance for a given body size is given as (predicted value of $T_{imm}$ under heat-hardening−predicted value of $T_{imm}$ acclimated to 17°C)/(predicted value of $T_{imm}$ acclimated to 28°C−predicted value of $T_{imm}$ acclimated to 17°C). All models were fitted with the same random effect structure with random intercept terms for beaker ($n = 16$ levels) and measuring run ($n = 23$ levels). Model assumptions were validated by visually inspecting residual heteroscedasticity and normality for both the most complex and best-fitting models tested. All statistical analyses were conducted in R version 3.5.2 [29].

## 3. Results

The models that included an interaction effect between body size and hardening treatment on heat tolerance provided the best fit to the data (models 1 and 2, table 1). However, adding treatment duration as a covariate yielded no further increase in the explanatory power of this relationship ($\Delta AICc = 1.59$, table 1).

This model revealed that the capacity for acclimation of $T_{imm}$ differed markedly with respect to body size in *D. magna*. The difference in $T_{imm}$ between individuals completely acclimated to the two control temperatures increased with body size, indicating that relatively old, large individuals have a greater capacity to acclimate than younger, smaller individuals (model estimate and data for 28°C control individuals versus 17°C control individuals, figure 3a and b, table 2). This resulted in different scaling relationships between heat tolerance and body size between the two temperatures to which the experimental subjects were initially acclimated. The slope of the relationship between $T_{imm}$ and body size was steeper

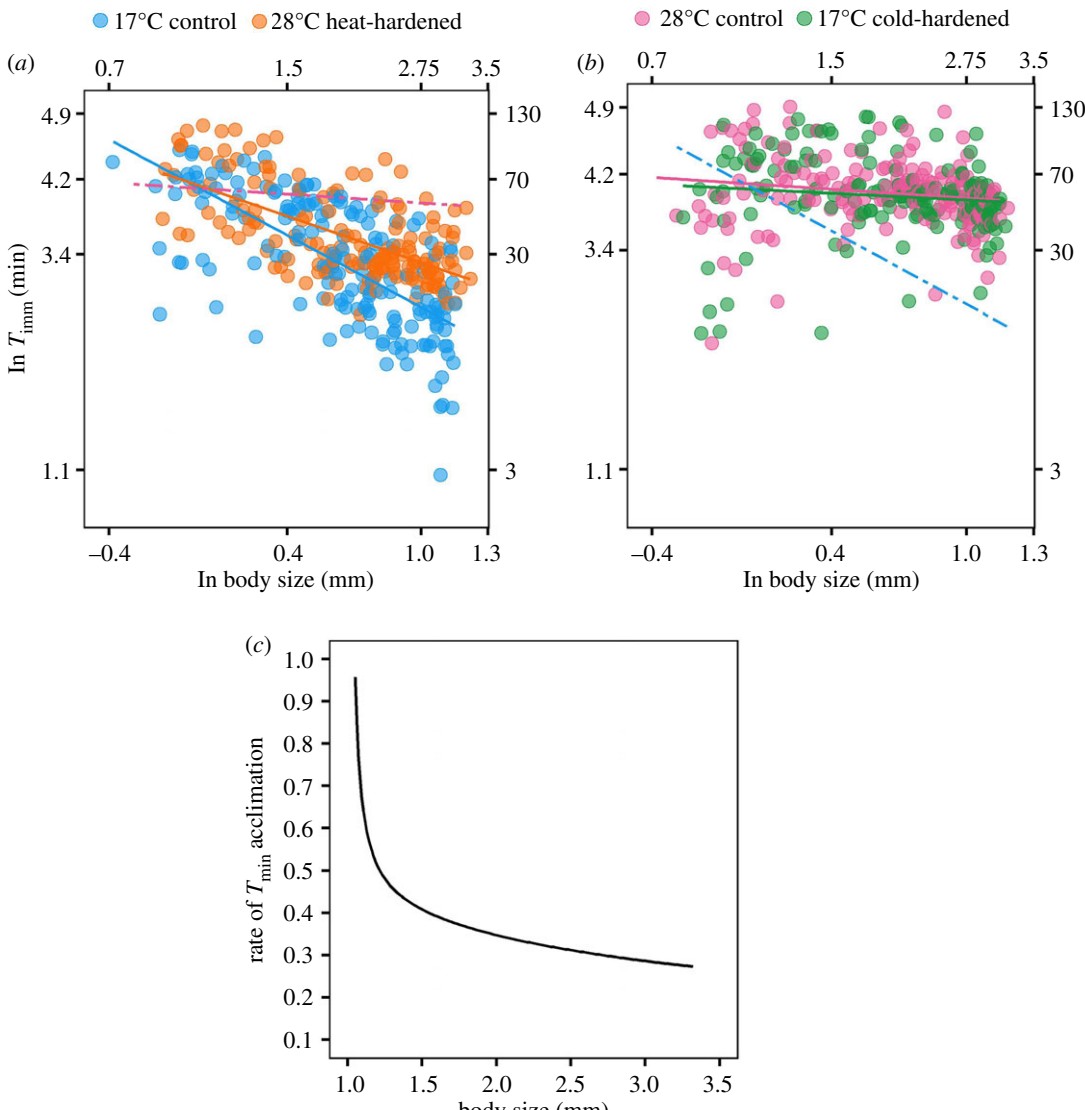

**Figure 3.** Relationship between $T_{imm}$ and body size (both plotted on natural logarithm scale) for *D. magna* acclimating to an increase (*a*) and decrease (*b*) in ambient temperature (28°C heat- and 17°C cold-hardened treatments in (*a*) and (*b*), respectively). In (*a*) and (*b*), data and model estimates (solid lines, colours correspond to symbols in the respective keys) for the hardened individuals are plotted relative to those for individuals from the corresponding control group. Dashed lines in each panel represent model estimates for the relationship between body size and $T_{imm}$ among individuals completely acclimated to each of the hardening temperatures (28°C and 17°C in (*a*) and (*b*), respectively). Secondary axes in each panel show un-transformed scale. Relationships on arithmetic scale are presented in the electronic supplementary material (electronic supplementary material, figure S2). $T_{imm}$ data are available from the Dryad Digital Repository: https://doi.org/10.5061/dryad.95x69p8g8 [26]. In (*c*), the relationship between the rate of acclimation in $T_{imm}$ and body size is shown for individuals acclimating to an increase in ambient temperature. See methods for description of calculation. Acclimation rate in response to a decrease in temperature was not calculated due to the lack of response to cold-hardening. (Online version in colour.)

**Table 1.** Set of candidate models testing the relationship between $T_{imm}$ of *Daphnia magna* in relation to hardening treatment (treatment), body size (size) and duration of the hardening treatment (treatment duration). ΔAIC$_c$: difference in AIC$_c$ values between a given model and the best-fitting model of those considered. $w_i$: probability that a given model is the best model of those considered. R$^2$: conditional r-squared for a given model, estimated using the MuMIn library in the R computing environment [30].

| model | fixed effects | *k* | AIC$_c$ | ΔAIC$_c$ | $w_i$ | acc $w_i$ | R$^2$ |
|---|---|---|---|---|---|---|---|
| 1 | treatment × size | 11 | 720.92 | 0.00 | 0.69 | 0.69 | 0.53 |
| 2 | treatment × size + treatment duration | 12 | 722.51 | 1.59 | 0.31 | 1.00 | 0.53 |
| 3 | treatment + size | 8 | 848.37 | 127.45 | 0.00 | 1.00 | 0.42 |
| 4 | treatment + size + treatment duration | 9 | 850.03 | 129.11 | 0.00 | 1.00 | 0.42 |

in animals acclimated to 17°C (slope ± s.e.; 17°C control individuals, −1.28 ± 0.08; 28°C control individuals −0.16 ± 0.08, figure 3*a* and *b*).

However, given that younger, smaller individuals showed less acclimation capacity than their larger counterparts, it was also evident that they were able to acclimate more rapidly

**Table 2.** Mixed effect model estimates describing relationships between body size and heat tolerance of *Daphnia magna* (both variables transformed to natural logarithm scale) from the control groups that were used to evaluate size-specific differences in acclimation capacity. The 17℃ control group is set as the intercept.

| term | estimate | s.e. | t-value | p-value |
|---|---|---|---|---|
| *acclimation capacity* | | | | |
| intercept (17℃ control) | 4.11 | 0.06 | 64.18 | <0.0001 |
| 28℃ control | −0.01 | 0.08 | −0.13 | 0.89 |
| ln body size | −1.28 | 0.08 | −15.54 | <0.0001 |
| 28℃ control × ln body size | 1.13 | 0.11 | 10.05 | <0.0001 |

(i.e. relatively small individuals obtain or come closer to obtaining complete acclimation following the brief period of heat-hardening, figure 3*c*, table 3). By contrast, we did not observe any change in heat tolerance among 17℃ cold-hardened individuals relative to individuals completely acclimated to 28°C (figure 3*b*, table 3). As such, we were unable to infer anything about the relationship between acclimation rate and body size in response to a decrease in temperature. This observation nonetheless shows that acquiring heat tolerance under heat-hardening proceeds at a faster rate than it does to lose it under cold-hardening.

## 4. Discussion

By measuring the heat tolerance of clonal individuals that differed in size and were either (i) completely acclimated to a relatively warm or cool temperature or (ii) in the process of acclimating to those same temperatures (i.e. hardening, after developing under the opposing thermal regime), we were able to infer how the rate of—and capacity for—acclimation changes throughout life in the zooplankton species *D. magna*. We observed that individuals from opposite ends of the ontogenetic spectrum displayed contrasting strategies in their tolerance of environmental stress and how this was modulated by acclimation to environmental change. Early in life, individuals had relatively high innate tolerance of thermal stress, irrespective of the temperature at which they (and their ancestors) had developed. Moreover, when their environment changed, the broad tolerance among juveniles was associated with a relatively minor capacity for acclimation, that in likely consequence meant they were able to acclimate heat tolerance rapidly. However, we observed that this pattern in innate heat tolerance and acclimation changed over the course of ontogeny. The initially robust, yet relatively inflexible juvenile phenotype, lost innate tolerance of thermal stress while simultaneously gaining capacity to increase heat tolerance by acclimating to temperature change (but at a slower rate than smaller, younger individuals). In seeking an explanation for the relationship between ontogeny, environmental tolerance and acclimation observed here, a recent hypothesis posits that juveniles may be incompletely acclimated to their birth environment, even when their parents (and grandparents) experienced the exact same set of conditions. As ontogeny

**Table 3.** Mixed effect model estimates describing relationships between body size and heat tolerance of *D. magna* (both variables transformed to natural logarithm scale) from the hardened and control groups that were used to evaluate size-specific differences in the rate of acclimation. The relationship between size and heat tolerance of the hardened groups is presented relative to the same relationship for the 17℃ and 28℃ control groups. Each comparison is presented with the hardened treatment group set as the intercept.

| term | estimate | s.e. | t-value | p-value |
|---|---|---|---|---|
| *increase in ambient temperature* | | | | |
| intercept (28℃ heat-hardened) | 4.13 | 0.07 | 59.97 | <0.0001 |
| 17℃ control | −0.02 | 0.09 | −0.23 | 0.82 |
| 28℃ control | −0.03 | 0.09 | −0.37 | 0.71 |
| ln body size | −0.82 | 0.09 | −9.63 | <0.0001 |
| 17℃ control × ln body size | −0.46 | 0.12 | −3.89 | <0.001 |
| 28℃ control × ln body size | 0.67 | 0.11 | 5.84 | <0.0001 |
| *decrease in ambient temperature* | | | | |
| intercept (17℃ cold-hardened) | 4.05 | 0.06 | 63.32 | <0.0001 |
| 17℃ control | 0.07 | 0.08 | 0.83 | 0.41 |
| 28℃ control | 0.06 | 0.08 | 0.71 | 0.48 |
| ln body size | −0.12 | 0.08 | −1.53 | 0.13 |
| 17℃ control × ln body size | −1.16 | 0.11 | −10.36 | <0.0001 |
| 28℃ control × ln body size | −0.04 | 0.11 | −0.35 | 0.73 |

proceeds, it is envisioned that the phenotype is then fine tuned to the prevailing environment experienced by the individual [31]. Here, this process may be evident as the transition from a tolerant, yet relatively rigid (but rapidly acclimating) juvenile phenotype to a less tolerant but more plastic (albeit slower to acclimate) adult phenotype. Mechanistic support for this hypothesis can be found in the accumulation of DNA methylation (a proposed key avenue by which the environment shapes the developing phenotype) with age in *D. magna* [32]. Although the latter study did not consider embryos or early juvenile life stages, genome wide methylation of developing avian embryos sampled from natural nests located across a gradient of habitat types was negligible in comparison to that of nestlings sampled only a short time later from the same nests [33]. However, we caution that the generality of this hypothesis requires further consideration. For instance, it may have greater utility in species where ontogeny is not associated with major changes in form, behaviour or habitat usage. In the current study, barring the onset of sexual maturation and production of young, the smallest and largest individuals measured are behaviourally and morphologically identical. Furthermore, the shallow water bodies inhabited by *D. magna* tend to offer little in the way of spatial variation in micro-habitat. However, in species that undergo distinct phenotypic shifts across development (e.g. from one

life stage to another), individuals of different size/life stage may have different sensitivities to the environment. For example, life stages with limited mobility might be expected to have a much larger capacity to acclimate to temperature change than life stages that can disperse in pursuit of more preferential temperatures [18]. Alternative explanations that can describe the scaling of the acclimation response observed here are elusive. For example, one might attempt to attribute the ontogenetic pattern in acclimation to a reduction in the innate heat tolerance of old, large individuals acclimated to 17°C. This explanation is consistent with the frequently observed decline in innate heat tolerance among older individuals [34–36]. However, such an explanation is still unable to account for the absence of acclimation in heat tolerance that we observed among relatively young, small individuals sourced from cultures maintained at markedly different temperatures.

Our results also concur with previous studies indicating that the rate of acclimation is asymmetric with respect to the direction of temperature change, proceeding more rapidly when temperature increases [10,12]. We also observed a shallower scaling relationship between heat tolerance and body size at higher acclimation temperatures. This negative relationship between the scaling of physiological traits and temperature is consistent with the pattern often observed in ectotherms [31,37–39]. Based on acute measurements of heat tolerance, several recent studies have suggested that older, larger individuals are likely to be the least resistant to extreme temperature events [23,40–42]. In these studies, heat tolerance was measured on individuals who had experienced a moderate temperature prior to measurement and thus had not had the possibility to acclimate to the higher than average temperatures that should precede extreme

heat episodes in nature. The shallower size scaling of heat tolerance among individuals acclimated to 28°C observed in the present study suggests that if organisms are sufficiently acclimated prior to heat stress (28°C is the uppermost temperature at which the experimental genotype can reproduce normally, [43]), there may be little difference in acute heat tolerance among individuals located at the opposing ends of the size and developmental spectrum.

In conclusion, our results provide evidence that there are substantial differences among individuals of the same species in how rapidly and by how much they can respond to environmental change. However, we wish to underscore that the data presented here represent a single phenotypic trait and a single species. More detailed investigation of the relationship between size/age and acclimation in other traits and species is required along with further consideration of the factors, both proximate and ultimate, that can explain these patterns of phenotypic variation.

Data accessibility. Our R code and data are available from the Dryad Digital Repository: https://doi.org/10.5061/dryad.95x69p8g8 [26].

Authors' contributions. T.B. conceived the idea for study and all authors participated in its design and execution. T.B. analysed the data and drafted the manuscript with input from H.-K.L. and S.E. All authors gave final approval for publication and agree to be held accountable for the work performed therein.

Competing interests. We declare we have no competing interests.

Funding. This work was supported by an H2020 Marie Skłodowska-Curie Actions International Fellowship (grant no. MSCA-IF 658530) and funding from the Research Council of Norway (Klimaforsk 244046, Centre of Excellence 223257/F50).

Acknowledgements. This work benefitted from the insightful comments of two anonymous referees.

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
