## [Reviewer comments · Proceedings of the Royal Society B: Biological Sciences]

Review History

RSPB-2019-2651.R0 (Original submission)

Review form: Reviewer 1

Recommendation

Major revision is needed (please make suggestions in comments)

Scientific importance: Is the manuscript an original and important contribution to its field?

Acceptable

General interest: Is the paper of sufficient general interest?

Good

Quality of the paper: Is the overall quality of the paper suitable?

Marginal

Is the length of the paper justified?

No

Should the paper be seen by a specialist statistical reviewer?

No

Do you have any concerns about statistical analyses in this paper? If so, please specify them explicitly in your report.

No

It is a condition of publication that authors make their supporting data, code and materials available - either as supplementary material or hosted in an external repository. Please rate, if applicable, the supporting data on the following criteria.

Is it accessible?

Yes

Is it clear?

No

Is it adequate?

No

Do you have any ethical concerns with this paper?

No

Comments to the Author

Comments for the authors:

This manuscript examines the effects of body size on the capacity for (and rate of) acclimation in upper thermal tolerance in *Daphnia magna*.

The authors find a strong negative effect of body size on upper thermal tolerance in animals long-term acclimated (4 generations) to 17C, but little or no effect of body size on upper thermal tolerance in animals acclimated to 28C, which all had relatively high tolerance. As a result, larger individuals showed greater capacity for acclimation, whereas smaller individuals maintained high tolerance under all conditions.

This is a very interesting observation that draws on the strengths of this study, which lie in the use of the *Daphnia magna* system. This allowed the use of a single clonal lineage, thus largely removing the effects of genetic variation among individuals. In addition, it was possible to perform the experiments with a large sample size, increasing the power of the inference. This makes this data set quite valuable relative to others that are available. However, the paper does only contain this one type of data, and thus the paper really only contains a single two-panel figure of results.

One challenge in interpreting the results is that there is an unavoidable confound between size and age. The fact that *Daphnia* are direct developers somewhat mitigates this problem in that at least there are no major changes in body morphology with age, but nevertheless it remains impossible to distinguish the effects of body size from the effects of age with this experimental design. As a result, making the connection between these data and the macrophysiological patterns that have been detected with respect to differences in acclimation capacity and rate among species is a bit challenging. This makes some of the discussion material a bit tenuous.

The weakest part of the study was the examination of the rate of acclimation. To examine the rate of acclimation, animals that were long-term acclimated to each of the experimental temperatures were "heat hardened" or "cold hardened" by ~10 days of exposure to the opposite temperature, and upper thermal tolerance was measured after hardening. Cold hardening over this time period had no effect on upper thermal tolerance, while heat hardening resulted in a partial shift towards the fully warm-acclimated phenotype. The authors conclude that large animals demonstrate a greater rate of acclimation than do smaller animals. However, I have three key concerns that make me wonder whether this conclusion is fully supported by the data.

My first, and perhaps least critical concern is that the experimental design examines only a single time point during acclimation (~10 days of either heat or cold hardening). To get an accurate estimate of rate, it would be substantially better to include several time points during the process of acclimation, as I see no reason to assume that the rate of acclimation is linear across the entire acclimation period. Nevertheless, at least some estimate of rate can be derived from these data.

My second, and somewhat more important concern has to do with the fact that there is a very large difference in the capacity for acclimation between large and small animals. It seems to me that this might result in a confound with respect to determining the rate of acclimation. As far as I can tell, in this paper, the rate of acclimation appears to be defined as the extent of phenotypic change within the ~10 day hardening period (although it is difficult to find a really clear statement of this definition anywhere in the paper). This definition is based on one from a previously published paper (Rohr et al. 2018). This definition works without difficulty for the conceptual example shown in Figure 1, but I am concerned that it may not be the most appropriate in the case of the data shown in Figure 3. Since small animals essentially do not acclimate, they have no ability to show a rate of acclimation, and I wonder whether this becomes an issue. Perhaps it would be interesting to think about the rate of acclimation in another way, perhaps in terms of the percent of the “fully acclimated” phenotype achieved during the hardening period. Would this change the conclusion?

My most important concern has to do with the fact that the data have been natural log transformed. It is not clear to me that this is justified. Certainly, the authors provide no such justification, and I struggle to think of one that is reasonable on mechanistic grounds. Nor it is clear to me that this transformation is necessary to linearize the data. At very least, the untransformed data should be available for examination in a supplemental Figure, because it seems to me that the absolute time of resistance is the ecologically important variable, rather than the ln transform of this time period. Examination of ln transformed data can be very deceptive, and I think this may be a particular problem with respect to the inferences about the rate of acclimation.

The authors should seriously consider looking at the untransformed data, or looking at the relationship between ln body mass and untransformed Timm.

I didn't run the raw data through the provided r code. Instead, I made a rough estimate based of the untransformed results from the data presented in Figure 3. It seems that small animals (with a body size of ~1.7mm) change their Timm from ~30 min to ~40 min with hardening (while fully warm-acclimated animals have a Timm of ~55min). Large animals (with a body size of approximately 3.1 mm) change their Timm from ~13.4 min to ~24 min (while fully warm-acclimated animals have a Timm of ~50 min). Thus, the absolute change in phenotype with heat hardening seems like it might actually be pretty similar between small and large animals. This suggests to me that the actual rate of change is not that different, even using the definition of rate adopted by the authors.

Since this was a bit of back of the envelope estimate, I can't say how true this conclusion may be, but I think it is critically important that the authors carefully examine their raw data to see whether this is actually the case.

In addition, if we ask the question of the extent to which the heat-hardened animals achieve the fully acclimated state, the small animals actually get much closer to the final state than do the larger animals. Taking this perspective could completely change the conclusions drawn.

Specific comments

Line 68-69: It is not entirely clear what is meant by developmental variation in the following sentence “because they develop directly after hatching, meaning that differences in size are not confounded with developmental variation”. I assume that you mean to imply differences among instars or other clearly morphologically different developmental stages, but I could easily imagine that there could be substantial metabolic or biochemical differences over time during

development in otherwise morphologically similar individuals, so I am not sure why having a direct developer necessarily controls for this issue. Indeed, at line 71-72 it is stated that mass-specific metabolic rate and heat tolerance vary with body size, which suggests that there is a direct effect of developmental stage on the physiology of these organisms. So I think this statement needs to be clarified or expanded upon a bit. Ultimately, direct development does not allow you to disentangle the effects of size from the effects of age, unless you have different clones that have different growth rates.

Line 73: The cited paper is under review and not available to the referee. What is important here is the rate of acclimation relative to the rate of change in size during growth. This is impossible for a referee to assess without access to these two pieces of information.

Line 95: The length of this fixed period seems like it would be critical, as different results might be obtained with different lengths of time (particularly if rates of acclimation are non linear). We are only told at Line 134 – 9-10 hours was chosen. How was this length of time determined? In particular, cold acclimation and warm acclimation are known to proceed at different rates in a variety of taxa. Why was the same amount of time chosen for both the heat hardening and the cold hardening experiments?

Line 151: L space missing between the word “stereomicroscope” and the word “as”

Line 198: What is known about the effect of photoperiod on heat tolerance in *Daphnia*? Does the fact that these animals were held at 24h light reduce any photoperiod effects?

Line 208: It is not clear to me that log transformation is necessary/appropriate. This choice needs to be justified (see my general comments) Also, I think it is important to be clear that you actually used natural log transformation, not log base 10. Again, this choice should be justified. There is substantial potential for deceiving yourself with respect to the patterns when looking at the log transformed data (particularly as presented in Figure 3 because high values are “squished” together on the y-axis, while low values are spread out, which can be a bit deceptive visually.

Line 224: I am not clear that Table 1 is necessary

Line 256-257: I disagree with the interpretation of the results presented here: “it was relatively large individuals who were able to increase Timm more than small individuals, indicating a faster rate of acclimation in heat”. I do not think the data necessarily support this interpretation of the data. See my general comments.

Line 269: The axes on Figure 3 should be clearly labelled to indicate that this is a natural logarithm (either by using log base e notation or ln).

Line 279: I think the untransformed data should be plotted in the supplemental material, and the data repository in FigShare should probably be referenced in the legend. I would also like to see an additional data supplement that includes the calculated Timm, so that the reader does not necessarily need to start with the raw data, but can simply examine the Timm data that you used to generate this figure and the untransformed data.

Line 308: “fine-tune” should be “fine-tunes”

Review form: Reviewer 2

Recommendation

Major revision is needed (please make suggestions in comments)

Scientific importance: Is the manuscript an original and important contribution to its field?

Good

General interest: Is the paper of sufficient general interest?

Good

Quality of the paper: Is the overall quality of the paper suitable?

Marginal

Is the length of the paper justified?

Yes

Should the paper be seen by a specialist statistical reviewer?

No

Do you have any concerns about statistical analyses in this paper? If so, please specify them explicitly in your report.

No

It is a condition of publication that authors make their supporting data, code and materials available - either as supplementary material or hosted in an external repository. Please rate, if applicable, the supporting data on the following criteria.

Is it accessible?

Yes

Is it clear?

Yes

Is it adequate?

Yes

Do you have any ethical concerns with this paper?

No

Comments to the Author

This is a potentially publishable manuscript that shows that acclimation rates and capacities vary within a species of clonal zooplankton with age/development/body size. The manuscript is well-written and the research appears to be well done. Furthermore the question is important and interesting.

However, the problem with the manuscript is that the authors completely confound age with body size and thus cannot conclude that body size is the driver. The authors finally acknowledge this in the Discussion but the entire framing of the paper and the abstract and title are based strictly on body size with no mention of development/age. Ironically, most of the Discussion focuses on development/age rather than body size, despite development/age not being mentioned before the Discussion. If the capacity for phenotypic plasticity develops with age, much like other physiological capacities, such as immunity, then it is not surprising that younger, smaller individuals acclimate less and at a slower rate. The authors also bring up other development/age hypothesis in the Discussion. In the absence of evidence against this being a developmental response, I simply cannot condone such a strong emphasis being placed on body size throughout the manuscript because that data do not support this emphasis. Thus, the Title, Abstract and Introduction would need to be revised so they address these three hypotheses (age, development, body size) in a more balanced manner.

One possible way around age and body size being confounded would be to attempt to decouple them experimentally, but this can be difficult and might not even be possible. The authors could attempt to do this by feeding clones that hatch at the same time different amounts of food, but I suspect that the ones that get more food will develop more quickly despite being of the same age as the other group. Nevertheless, this could help address this issue.

Title: The title is a bit misleading. It gives the impression of generality, but the study is only on intraspecific variation within a single species. The title should be revised to make clear the narrow scope of the study and to reduce the emphasis on body size only.

L 337-341: Yes, but a crucial distinction between the inter- and intraspecific comparisons is that the former does not confound age/development with body size and the latter does. So, comparing the two is like comparing apples with oranges. Making this clear here is very important.

L 341-343: You cannot conclude that mass-specific metabolic rate is not a driver of the interspecific comparisons if all or most of the intraspecific comparisons are confounded. Please revise this sentence accordingly.

L 345-349: Again, you have no evidence to support the notion "that different mechanisms may be driving the relationships between acclimation rate and body size observed at each level of organization." Almost all of the intraspecific studies confound age/development with body size. So, differences between inter- and intraspecific studies are just as likely to be due to this confounder as true underlying differences in mechanisms. Please address.

L 372-376: I appreciate this ending to the manuscript.

Decision letter (RSPB-2019-2651.R0)

04-Dec-2019

Dear Dr Burton:

I am writing to inform you that your manuscript RSPB-2019-2651 entitled "Small individuals acclimate less and at a slower rate" has, in its current form, been rejected for publication in Proceedings B.

This action has been taken on the advice of referees, who have recommended that substantial revisions are necessary. With this in mind we would be happy to consider a resubmission, provided the comments of the referees are fully addressed. However please note that this is not a provisional acceptance.

- 1) A 'response to referees' document including details of how you have responded to the comments, and the adjustments you have made.
- 2) A clean copy of the manuscript and one with 'tracked changes' indicating your 'response to referees' comments document.

3) Line numbers in your main document.

Please note that this decision may (or may not) have taken into account confidential comments.

In your revision process, please take a second look at how open your science is; our policy is that all data involved with the study should be made openly accessible-- see: <https://royalsociety.org/journals/ethics-policies/data-sharing-mining/>
Insufficient sharing of data can delay or even cause rejection of a paper.

Sincerely,
Professor John Hutchinson, Editor
mailto: proceedingsb@royalsociety.org

Associate Editor
Comments to Author:

We have now obtained two reviews of your manuscript from experts in the field. Both reviewers found the topic to be interesting and the choice of study species appropriate. With that said, they also noted an important confound in the data between age and size, which makes it difficult to attribute the findings to one or the other factor. This issue manifests as a shift from focusing on body size early in the manuscript to development/age in the discussion. One reviewer raises substantial concerns with the data analysis and whether it is appropriate to do a natural log transformation on the data, and if the definition of the rate of acclimation used in the paper is appropriate. The suggestion to include the untransformed data is appropriate.

Reviewer(s)' Comments to Author:

Referee: 1

Comments to the Author(s)
Comments for the authors:

This manuscript examines the effects of body size on the capacity for (and rate of) acclimation in upper thermal tolerance in *Daphnia magna*.

The authors find a strong negative effect of body size on upper thermal tolerance in animals long-term acclimated (4 generations) to 17C, but little or no effect of body size on upper thermal tolerance in animals acclimated to 28C, which all had relatively high tolerance. As a result, larger individuals showed greater capacity for acclimation, whereas smaller individuals maintained high tolerance under all conditions.

This is a very interesting observation that draws on the strengths of this study, which lie in the use of the *Daphnia magna* system. This allowed the use of a single clonal lineage, thus largely removing the effects of genetic variation among individuals. In addition, it was possible to perform the experiments with a large sample size, increasing the power of the inference. This makes this data set quite valuable relative to others that are available. However, the paper does only contain this one type of data, and thus the paper really only contains a single two-panel figure of results.

One challenge in interpreting the results is that there is an unavoidable confound between size and age. The fact that *Daphnia* are direct developers somewhat mitigates this problem in that at

least there are no major changes in body morphology with age, but nevertheless it remains impossible to distinguish the effects of body size from the effects of age with this experimental design. As a result, making the connection between these data and the macrophysiological patterns that have been detected with respect to differences in acclimation capacity and rate among species is a bit challenging. This makes some of the discussion material a bit tenuous.

The weakest part of the study was the examination of the rate of acclimation. To examine the rate of acclimation, animals that were long-term acclimated to each of the experimental temperatures were “heat hardened” or “cold hardened” by ~10 days of exposure to the opposite temperature, and upper thermal tolerance was measured after hardening. Cold hardening over this time period had no effect on upper thermal tolerance, while heat hardening resulted in a partial shift towards the fully warm-acclimated phenotype. The authors conclude that large animals demonstrate a greater rate of acclimation than do smaller animals. However, I have three key concerns that make me wonder whether this conclusion is fully supported by the data.

My first, and perhaps least critical concern is that the experimental design examines only a single time point during acclimation (~10 days of either heat or cold hardening). To get an accurate estimate of rate, it would be substantially better to include several time points during the process of acclimation, as I see no reason to assume that the rate of acclimation is linear across the entire acclimation period. Nevertheless, at least some estimate of rate can be derived from these data.

My second, and somewhat more important concern has to do with the fact that there is a very large difference in the capacity for acclimation between large and small animals. It seems to me that this might result in a confound with respect to determining the rate of acclimation. As far as I can tell, in this paper, the rate of acclimation appears to be defined as the extent of phenotypic change within the ~10 day hardening period (although it is difficult to find a really clear statement of this definition anywhere in the paper). This definition is based on one from a previously published paper (Rohr et al. 2018). This definition works without difficulty for the conceptual example shown in Figure 1, but I am concerned that it may not be the most appropriate in the case of the data shown in Figure 3. Since small animals essentially do not acclimate, they have no ability to show a rate of acclimation, and I wonder whether this becomes an issue. Perhaps it would be interesting to think about the rate of acclimation in another way, perhaps in terms of the percent of the “fully acclimated” phenotype achieved during the hardening period. Would this change the conclusion?

My most important concern has to do with the fact that the data have been natural log transformed. It is not clear to me that this is justified. Certainly, the authors provide no such justification, and I struggle to think of one that is reasonable on mechanistic grounds. Nor it is clear to me that this transformation is necessary to linearize the data. At very least, the untransformed data should be available for examination in a supplemental Figure, because it seems to me that the absolute time of resistance is the ecologically important variable, rather than the ln transform of this time period. Examination of ln transformed data can be very deceptive, and I think this may be a particular problem with respect to the inferences about the rate of acclimation.

The authors should seriously consider looking at the untransformed data, or looking at the relationship between ln body mass and untransformed Timm.

I didn't run the raw data through the provided r code. Instead, I made a rough estimate based of the untransformed results from the data presented in Figure 3. It seems that small animals (with a body size of ~1.7mm) change their Timm from ~30 min to ~40 min with hardening (while fully warm-acclimated animals have a Timm of ~55min). Large animals (with a body size of approximately 3.1 mm) change their Timm from ~13.4 min to ~24 min (while fully warm-acclimated animals have a Timm of ~50 min). Thus, the absolute change in phenotype with heat hardening seems like it might actually be pretty similar between small and large animals. This suggests to me that the actual rate of change is not that different, even using the definition of rate adopted by the authors.

Since this was a bit of back of the envelope estimate, I can't say how true this conclusion may be, but I think it is critically important that the authors carefully examine their raw data to see whether this is actually the case.

In addition, if we ask the question of the extent to which the heat-hardened animals achieve the fully acclimated state, the small animals actually get much closer to the final state than do the larger animals. Taking this perspective could completely change the conclusions drawn.

Specific comments

Line 68-69: It is not entirely clear what is meant by developmental variation in the following sentence "because they develop directly after hatching, meaning that differences in size are not confounded with developmental variation". I assume that you mean to imply differences among instars or other clearly morphologically different developmental stages, but I could easily imagine that there could be substantial metabolic or biochemical differences over time during development in otherwise morphologically similar individuals, so I am not sure why having a direct developer necessarily controls for this issue. Indeed, at line 71-72 it is stated that mass-specific metabolic rate and heat tolerance vary with body size, which suggests that there is a direct effect of developmental stage on the physiology of these organisms. So I think this statement needs to be clarified or expanded upon a bit. Ultimately, direct development does not allow you to disentangle the effects of size from the effects of age, unless you have different clones that have different growth rates.

Line 73: The cited paper is under review and not available to the referee. What is important here is the rate of acclimation relative to the rate of change in size during growth. This is impossible for a referee to assess without access to these two pieces of information.

Line 95: The length of this fixed period seems like it would be critical, as different results might be obtained with different lengths of time (particularly if rates of acclimation are non linear). We are only told at Line 134 - 9-10 hours was chosen. How was this length of time determined? In particular, cold acclimation and warm acclimation are known to proceed at different rates in a variety of taxa. Why was the same amount of time chosen for both the heat hardening and the cold hardening experiments?

Line 151: L space missing between the word "stereomicroscope" and the word "as"

Line 198: What is known about the effect of photoperiod on heat tolerance in *Daphnia*? Does the fact that these animals were held at 24h light reduce any photoperiod effects?

Line 208: It is not clear to me that log transformation is necessary/appropriate. This choice needs to be justified (see my general comments) Also, I think it is important to be clear that you actually used natural log transformation, not log base 10. Again, this choice should be justified. There is substantial potential for deceiving yourself with respect to the patterns when looking at the log transformed data (particularly as presented in Figure 3 because high values are "squished" together on the y-axis, while low values are spread out, which can be a bit deceptive visually.

Line 224: I am not clear that Table 1 is necessary

Line 256-257: I disagree with the interpretation of the results presented here: "it was relatively large individuals who were able to increase Timm more than small individuals, indicating a faster rate of acclimation in heat". I do not think the data necessarily support this interpretation of the data. See my general comments.

Line 269: The axes on Figure 3 should be clearly labelled to indicate that this is a natural logarithm (either by using log base e notation or ln).

Line 279: I think the untransformed data should be plotted in the supplemental material, and the

data repository in FigShare should probably be referenced in the legend. I would also like to see an additional data supplement that includes the calculated Timm, so that the reader does not necessarily need to start with the raw data, but can simply examine the Timm data that you used to generate this figure and the untransformed data.

Line 308: “fine-tune” should be “fine-tunes”

Referee: 2

Comments to the Author(s)

This is a potentially publishable manuscript that shows that acclimation rates and capacities vary within a species of clonal zooplankton with age/development/body size. The manuscript is well-written and the research appears to be well done. Furthermore the question is important and interesting.

However, the problem with the manuscript is that the authors completely confound age with body size and thus cannot conclude that body size is the driver. The authors finally acknowledge this in the Discussion but the entire framing of the paper and the abstract and title are based strictly on body size with no mention of development/age. Ironically, most of the Discussion focuses on development/age rather than body size, despite development/age not being mentioned before the Discussion. If the capacity for phenotypic plasticity develops with age, much like other physiological capacities, such as immunity, then it is not surprising that younger, smaller individuals acclimate less and at a slower rate. The authors also bring up other development/age hypothesis in the Discussion. In the absence of evidence against this being a developmental response, I simply cannot condone such a strong emphasis being placed on body size throughout the manuscript because that data do not support this emphasis. Thus, the Title, Abstract and Introduction would need to be revised so they address these three hypotheses (age, development, body size) in a more balanced manner.

One possible way around age and body size being confounded would be to attempt to decouple them experimentally, but this can be difficult and might not even be possible. The authors could attempt to do this by feeding clones that hatch at the same time different amounts of food, but I suspect that the ones that get more food will develop more quickly despite being of the same age as the other group. Nevertheless, this could help address this issue.

Title: The title is a bit misleading. It gives the impression of generality, but the study is only on intraspecific variation within a single species. The title should be revised to make clear the narrow scope of the study and to reduce the emphasis on body size only.

L 337-341: Yes, but a crucial distinction between the inter- and intraspecific comparisons is that the former does not confound age/development with body size and the latter does. So, comparing the two is like comparing apples with oranges. Making this clear here is very important.

L 341-343: You cannot conclude that mass-specific metabolic rate is not a driver of the interspecific comparisons if all or most of the intraspecific comparisons are confounded. Please revise this sentence accordingly.

L 345-349: Again, you have no evidence to support the notion “that different mechanisms may be driving the relationships between acclimation rate and body size observed at each level of organization.” Almost all of the intraspecific studies confound age/development with body size. So, differences between inter- and intraspecific studies are just as likely to be due to this confounder as true underlying differences in mechanisms. Please address.

L 372-376: I appreciate this ending to the manuscript.

Author's Response to Decision Letter for (RSPB-2019-2651.R0)

See Appendix A.

RSPB-2020-0189.R0

Review form: Reviewer 1

Recommendation

Accept with minor revision (please list in comments)

Scientific importance: Is the manuscript an original and important contribution to its field?

Acceptable

General interest: Is the paper of sufficient general interest?

Good

Quality of the paper: Is the overall quality of the paper suitable?

Acceptable

Is the length of the paper justified?

Yes

Should the paper be seen by a specialist statistical reviewer?

No

Do you have any concerns about statistical analyses in this paper? If so, please specify them explicitly in your report.

No

It is a condition of publication that authors make their supporting data, code and materials available - either as supplementary material or hosted in an external repository. Please rate, if applicable, the supporting data on the following criteria.

Is it accessible?

Yes

Is it clear?

Yes

Is it adequate?

Yes

Do you have any ethical concerns with this paper?

No

Comments to the Author

General comments:

The paper is improved relative to the previous version, and I greatly appreciated the inclusion of the non-transformed data in the supplement, although I would have appreciated a bit more

analysis of the data in this form (e.g. a non linear fit to the data). The paper is generally very well written. The question addressed is interesting, and the work has been well performed, although the fundamental confound between size and age remains.

However, I still struggle with some of the interpretation of the data. Although I agree with the vast majority of the authors' interpretation of their data, I am not sure that I agree with the section of the discussion around lines 340-347. In this section of the discussion, the authors make the argument that younger/smaller animals may be viewed as generalists with the ability to rapidly respond to environmental change via fast acclimation, while larger/older individuals may be viewed as specialists that may be more susceptible to rapid environmental change. But to me this argument seems to entirely ignore the differences in acclimation capacity between the two life stages (discussed at line 313-316), since younger/smaller individuals show limited acclimation capacity and high overall hardiness and larger/older individuals show substantial acclimation capacity, but lower overall hardiness. I suppose this issue comes down to exactly what you mean by generalist vs specialist. Is a non-plastic but highly tolerant individual a generalist because it can cope with most of the environments it is likely to encounter? Or is a plastic but less tolerant individual a generalist because it can adopt multiple phenotypes? I think this actually highlights a logical flaw in setting this up as a binary choice. In fact, in much of the plasticity literature, I think that at least three alternative strategies are usually envisaged (generalist, specialist, and plastic). See for example, Phenotypic Plasticity: Functional and Conceptual Approaches, DeWhitt and Scheiner. While this seems to be a bit of a semantic issue, I think it actually is rather fundamental in terms of the way the data will be interpreted. Are young/small individuals likely to be able to cope with environmental change because they have the capacity to rapidly (but very marginally) change their phenotype, or simply because they are very hardy to start with? This makes a big difference in what you take away from the study.

I think this also impinges on the overall framing of the study in the introduction, which still has at least some focus on the previously observed inter-specific pattern of differences in rates of acclimation between large and small individuals (although this is somewhat reduced relative to the previous version). Unfortunately, I think this experiment has very little to say with respect to this question, as the patterns in *Daphnia* may simply reflect different strategies adopted at different life history stages, rather than by individuals of different sizes. Indeed, the observed patterns in the data seem more consistent with a life-history based hypothesis (i.e. producing young that are ready for anything, but having older individuals that can shape their phenotype through plasticity in response to the prevailing environment). Overall, I think the paper would be better framed as addressing whether acclimation capacity and/or rate varies across development than whether acclimation capacity and/or rate varies with body size, given the fundamental confound between these two issues with this experimental design.

Minor specific comments

Line 22-23: I prefer to see a somewhat more quantitative and detailed description of the results, rather than just the conclusions from them, particularly in a case where I think the interpretation of the results is somewhat open to question.

Line 38: I am not sure I agree with the statement "measurements of the rate at which [acclimation] progresses within-species are surprisingly rare". First, any single study of the rate of acclimation is necessarily a within-species examination, so I am not sure what this sentence is trying to say. Second, there are actually a rather large number of studies of the rate of acclimation of various traits (especially in fish). Perhaps this sentence was intended to read "measurements of variation in the rate at which acclimation progresses within species are surprisingly rare". This, I think, is a fairly true statement.

Line 107-109: I am not sure the logic of choosing 40% of the time for "full acclimation" is as solid as this statement implies. It seems to me to be very dependent on the reliability of your estimate of the time to full acclimation. For example, if you rely on data that were taken at only a single time point (after some longish period of acclimation), you have no idea how long prior to that

time point “full acclimation” was reached. As a result, you may be back calculating your 40% time point from the wrong spot. I think I must be missing something in the logical chain. Perhaps expand the justification of this choice a bit more.

Line 340-346: As mentioned in the general comments, I struggle with the logic of this section of the discussion.

Review form: Reviewer 2 (Jason R. Rohr)

Recommendation

Accept as is

Scientific importance: Is the manuscript an original and important contribution to its field?

Good

General interest: Is the paper of sufficient general interest?

Good

Quality of the paper: Is the overall quality of the paper suitable?

Good

Is the length of the paper justified?

Yes

Should the paper be seen by a specialist statistical reviewer?

No

Do you have any concerns about statistical analyses in this paper? If so, please specify them explicitly in your report.

No

It is a condition of publication that authors make their supporting data, code and materials available - either as supplementary material or hosted in an external repository. Please rate, if applicable, the supporting data on the following criteria.

Is it accessible?

Yes

Is it clear?

Yes

Is it adequate?

Yes

Do you have any ethical concerns with this paper?

No

Comments to the Author

The authors have adequately addressed my concerns. The manuscript is well-written and the research appears to be well done. Furthermore the question is important and interesting.

Decision letter (RSPB-2020-0189.R0)

12-Feb-2020

Dear Dr Burton:

Your manuscript has now been peer reviewed and the reviews have been assessed by an Associate Editor. The reviewers' comments (not including confidential comments to the Editor) and the comments from the Associate Editor are included at the end of this email for your reference. As you will see, the reviewers and the Editors have raised some concerns with your manuscript and we would like to invite you to revise your manuscript to address them.

We do not allow multiple rounds of revision so we urge you to make every effort to fully address all of the comments at this stage. If deemed necessary by the Associate Editor, your manuscript will be sent back to one or more of the original reviewers for assessment. If the original reviewers are not available we may invite new reviewers. Please note that we cannot guarantee eventual acceptance of your manuscript at this stage. 1 reviewer is very satisfied but the other has major lingering concerns about the analyses which must be overcome if the study is to be publishable.

Research ethics:

Use of animals and field studies:

It is a condition of publication that you make available the data and research materials supporting the results in the article. Datasets should be deposited in an appropriate publicly available repository and details of the associated accession number, link or DOI to the datasets must be included in the Data Accessibility section of the article

(<https://royalsociety.org/journals/ethics-policies/data-sharing-mining/>). Reference(s) to datasets should also be included in the reference list of the article with DOIs (where available).

[http://datadryad.org/submit?journalID=RSPB&manu=\(Document not available\)](http://datadryad.org/submit?journalID=RSPB&manu=(Document%20not%20available)), which will take you to your unique entry in the Dryad repository.

Please submit a copy of your revised paper within three weeks. If we do not hear from you within this time your manuscript will be rejected. If you are unable to meet this deadline please let us know as soon as possible, as we may be able to grant a short extension.

Best wishes,
Professor John Hutchinson, Editor
mailto: proceedingsb@royalsociety.org

Associate Editor Board Member

Comments to Author:

We have received feedback on your resubmission to Proceedings B. While one reviewer had no further comments for you to address, the other reviewer provided a number of points that were not fully addressed in this version. I would appreciate if you could please deal with these in a revised version, paying particular attention to the concerns regarding the interpretation of the data as being driven by large versus small individuals compared to differences in life history stages.

Reviewer(s)' Comments to Author:

Referee: 2

Comments to the Author(s).

The authors have adequately addressed my concerns. The manuscript is well-written and the research appears to be well done. Furthermore the question is important and interesting.

Referee: 1

Comments to the Author(s).

General comments:

The paper is improved relative to the previous version, and I greatly appreciated the inclusion of the non-transformed data in the supplement, although I would have appreciated a bit more analysis of the data in this form (e.g. a non linear fit to the data). The paper is generally very well written. The question addressed is interesting, and the work has been well performed, although the fundamental confound between size and age remains.

However, I still struggle with some of the interpretation of the data. Although I agree with the vast majority of the authors' interpretation of their data, I am not sure that I agree with the section of the discussion around lines 340-347. In this section of the discussion, the authors make the argument that younger/smaller animals may be viewed as generalists with the ability to rapidly respond to environmental change via fast acclimation, while larger/older individuals may be viewed as specialists that may be more susceptible to rapid environmental change. But to me this argument seems to entirely ignore the differences in acclimation capacity between the two life stages (discussed at line 313-316), since younger/smaller individuals show limited acclimation capacity and high overall hardiness and larger/older individuals show substantial acclimation capacity, but lower overall hardiness. I suppose this issue comes down to exactly what you mean by generalist vs specialist. Is a non-plastic but highly tolerant individual a generalist because it can cope with most of the environments it is likely to encounter? Or is a plastic but less tolerant individual a generalist because it can adopt multiple phenotypes? I think this actually highlights a logical flaw in setting this up as a binary choice. In fact, in much of the plasticity literature, I think that at least three alternative strategies are usually envisaged (generalist, specialist, and plastic). See for example, Phenotypic Plasticity: Functional and Conceptual Approaches, DeWhitt and Scheiner. While this seems to be a bit of a semantic issue, I think it actually is rather fundamental in terms of the way the data will be interpreted. Are young/small individuals likely to be able to cope with environmental change because they have the capacity to rapidly (but very marginally) change their phenotype, or simply because they are very hardy to start with? This makes a big difference in what you take away from the study.

I think this also impinges on the overall framing of the study in the introduction, which still has at least some focus on the previously observed inter-specific pattern of differences in rates of acclimation between large and small individuals (although this is somewhat reduced relative to the previous version). Unfortunately, I think this experiment has very little to say with respect to this question, as the patterns in *Daphnia* may simply reflect different strategies adopted at different life history stages, rather than by individuals of different sizes. Indeed, the observed patterns in the data seem more consistent with a life-history based hypothesis (i.e. producing young that are ready for anything, but having older individuals that can shape their phenotype through plasticity in response to the prevailing environment). Overall, I think the paper would be better framed as addressing whether acclimation capacity and/or rate varies across development than whether acclimation capacity and/or rate varies with body size, given the fundamental confound between these two issues with this experimental design.

Minor specific comments

Line 22-23: I prefer to see a somewhat more quantitative and detailed description of the results, rather than just the conclusions from them, particularly in a case where I think the interpretation of the results is somewhat open to question.

Line 38: I am not sure I agree with the statement "measurements of the rate at which [acclimation] progresses within-species are surprisingly rare". First, any single study of the rate of acclimation is necessarily a within-species examination, so I am not sure what this sentence is trying to say. Second, there are actually a rather large number of studies of the rate of acclimation of various traits (especially in fish). Perhaps this sentence was intended to read "measurements of

variation in the rate at which acclimation progresses within species are surprisingly rare". This, I think, is a fairly true statement.

Line 107-109: I am not sure the logic of choosing 40% of the time for "full acclimation" is as solid as this statement implies. It seems to me to be very dependent on the reliability of your estimate of the time to full acclimation. For example, if you rely on data that were taken at only a single time point (after some longish period of acclimation), you have no idea how long prior to that time point "full acclimation" was reached. As a result, you may be back calculating your 40% time point from the wrong spot. I think I must be missing something in the logical chain. Perhaps expand the justification of this choice a bit more.

Line 340-346: As mentioned in the general comments, I struggle with the logic of this section of the discussion.

Author's Response to Decision Letter for (RSPB-2020-0189.R0)

See Appendix B.

RSPB-2020-0189.R1 (Revision)

Review form: Reviewer 1

Recommendation

Accept as is

Scientific importance: Is the manuscript an original and important contribution to its field?

Excellent

General interest: Is the paper of sufficient general interest?

Excellent

Quality of the paper: Is the overall quality of the paper suitable?

Excellent

Is the length of the paper justified?

Yes

Should the paper be seen by a specialist statistical reviewer?

No

Do you have any concerns about statistical analyses in this paper? If so, please specify them explicitly in your report.

No

It is a condition of publication that authors make their supporting data, code and materials available - either as supplementary material or hosted in an external repository. Please rate, if applicable, the supporting data on the following criteria.

Is it accessible?

Yes

Is it clear?

Yes

Is it adequate?

Yes

Do you have any ethical concerns with this paper?

No

Comments to the Author

The authors have done an excellent job of addressing my remaining concerns.

Decision letter (RSPB-2020-0189.R1)

10-Mar-2020

Dear Dr Burton

I am pleased to inform you that your manuscript entitled "Acclimation capacity and rate change through life in the zooplankton *Daphnia*" has been accepted for publication in Proceedings B. Congratulations!!

Open Access

Your article has been estimated as being 8 pages long. Our Production Office will be able to confirm the exact length at proof stage.

Paper charges

All supplementary materials accompanying an accepted article will be treated as in their final form. They will be published alongside the paper on the journal website and posted on the online

figshare repository. Files on figshare will be made available approximately one week before the accompanying article so that the supplementary material can be attributed a unique DOI.

Sincerely,

Dr John Hutchinson

Associate Editor:

Board Member: 1

Comments to Author:

(There are no comments.)

Board Member: 2

Comments to Author:

(There are no comments.)

Appendix A

Dear Professor Hutchinson,

We thank both referees for their constructive feedback on our manuscript. We have revised the manuscript accordingly and address each of their comments (numbered) below. In each case, we list the original comment and follow it with our response in italics.

Associate Editor

Comments to Author:

We have now obtained two reviews of your manuscript from experts in the field. Both reviewers found the topic to be interesting and the choice of study species appropriate. With that said, they also noted an important confound in the data between age and size, which makes it difficult to attribute the findings to one or the other factor. This issue manifests as a shift from focusing on body size early in the manuscript to development/age in the discussion. One reviewer raises substantial concerns with the data analysis and whether it is appropriate to do a natural log transformation on the data, and if the definition of the rate of acclimation used in the paper is appropriate. The suggestion to include the untransformed data is appropriate.

Referee: 1

This manuscript examines the effects of body size on the capacity for (and rate of) acclimation in upper thermal tolerance in *Daphnia magna*.

The authors find a strong negative effect of body size on upper thermal tolerance in animals long-term acclimated (4 generations) to 17C, but little or no effect of body size on upper thermal tolerance in animals acclimated to 28C, which all had relatively high tolerance. As a result, larger individuals showed greater capacity for acclimation, whereas smaller individuals maintained high tolerance under all conditions.

This is a very interesting observation that draws on the strengths of this study, which lie in the use of the *Daphnia magna* system. This allowed the use of a single clonal lineage, thus largely removing the effects of genetic variation among individuals. In addition, it was possible to perform the experiments with a large sample size, increasing the power of the inference. This makes this data set quite valuable relative to others that are available. However, the paper does only contain this one type of data, and thus the paper really only contains a single two-panel figure of results.

1. One challenge in interpreting the results is that there is an unavoidable confound between size and age. The fact that *Daphnia* are direct developers somewhat mitigates this problem in that at least there are no major changes in body morphology with age, but nevertheless it remains impossible to distinguish the effects of body size from the effects of age with this experimental design. As a result, making the connection between these data and the macrophysiological patterns that have been detected with respect to differences in acclimation capacity and rate among species is a bit challenging. This makes some of the discussion material a bit tenuous.

See response to Comment 1 by Referee 2 regarding separation of size effects vs age effects on acclimation. Regarding the comparison between our data and macrophysiological/inter-specific patterns see response to comments 4, 5 and 6 by Referee 2.

2. The weakest part of the study was the examination of the rate of acclimation. To examine the rate of acclimation, animals that were long-term acclimated to each of the experimental temperatures

were “heat hardened” or “cold hardened” by ~10 days of exposure to the opposite temperature, and upper thermal tolerance was measured after hardening. Cold hardening over this time period had no effect on upper thermal tolerance, while heat hardening resulted in a partial shift towards the fully warm-acclimated phenotype. The authors conclude that large animals demonstrate a greater rate of acclimation than do smaller animals. However, I have three key concerns that make me wonder whether this conclusion is fully supported by the data.

See separate responses to comments 3, 4 and 5 below.

3. My first, and perhaps least critical concern is that the experimental design examines only a single time point during acclimation (~10 days of either heat or cold hardening). To get an accurate estimate of rate, it would be substantially better to include several time points during the process of acclimation, as I see no reason to assume that the rate of acclimation is linear across the entire acclimation period. Nevertheless, at least some estimate of rate can be derived from these data.

First, wish to clarify that the acclimation period was ~10 hours, not days (see Fig. 2). We have now added a rationale for this choice (line 107). In principle, we agree with Referee 1 that measurements made at multiple time points would aid in quantifying any non-linearity in the rate of acclimation. However, given the paucity of data on this subject, our primary goal was simply to determine if individuals of different age/size show variation in how they acclimate. Moreover, as pointed out in the next comment, we were surprised to observe that small individuals did not actually acclimate at all. I.e. T_{imm} was similar for small individuals independent of their prior temperature experience (whole life and parental generations) at 28 degrees vs 17 degrees vs 10 h heat hardening, see Fig. 3A). Thus, measurement of heat tolerance following shorter or longer heat hardening would not have revealed anything different for these individuals.

4. My second, and somewhat more important concern has to do with the fact that there is a very large difference in the capacity for acclimation between large and small animals. It seems to me that this might result in a confound with respect to determining the rate of acclimation. As far as I can tell, in this paper, the rate of acclimation appears to be defined as the extent of phenotypic change within the ~10 day hardening period (although it is difficult to find a really clear statement of this definition anywhere in the paper). This definition is based on one from a previously published paper (Rohr et al. 2018). This definition works without difficulty for the conceptual example shown in Figure 1, but I am concerned that it may not be the most appropriate in the case of the data shown in Figure 3. Since small animals essentially do not acclimate, they have no ability to show a rate of acclimation, and I wonder whether this becomes an issue. Perhaps it would be interesting to think about the rate of acclimation in another way, perhaps in terms of the percent of the “fully acclimated” phenotype achieved during the hardening period. Would this change the conclusion?

We thank Referee 1 for this excellent suggestion and have adopted their suggested definition of acclimation rate. We now outline this in the Introduction (lines ca. 43 - 53) and in the caption for Figure 1 (along with definitions for acclimation capacity in the same places). As a result of the changed definition we have also adjusted text in the Methods describing how acclimation rate was calculated (lines ca. 234 – 243). Given the new definition of acclimation rate, we have included an additional panel in Figure 3 showing how this trait changes with body size in response to heat-hardening.

5. My most important concern has to do with the fact that the data have been natural log transformed. It is not clear to me that this is justified. Certainly, the authors provide no such justification, and I struggle to think of one that is reasonable on mechanistic grounds. Nor it is clear to me that this transformation is necessary to linearize the data. At very least, the

untransformed data should be available for examination in a supplemental Figure, because it seems to me that the absolute time of resistance is the ecologically important variable, rather than the ln transform of this time period. Examination of ln transformed data can be very deceptive, and I think this may be a particular problem with respect to the inferences about the rate of acclimation.

We have added text to Methods describing the two reasons was adopted log-log transformation of our data. (1) so that variation in acclimation could be examined proportionately with respect to differences in body size and (2) to satisfy assumptions of linear modelling (lines ca. 222- 224). We have included an extra supplementary figure (S2) showing the same data on arithmetic scale.

6. The authors should seriously consider looking at the untransformed data, or looking at the relationship between ln body mass and untransformed Timm. I didn't run the raw data through the provided r code. Instead, I made a rough estimate based of the untransformed results from the data presented in Figure 3. It seems that small animals (with a body size of ~1.7mm) change their Timm from ~30 min to ~40 min with hardening (while fully warm-acclimated animals have a Timm of ~55min). Large animals (with a body size of approximately 3.1 mm) change their Timm from ~13.4 min to ~24 min (while fully warm-acclimated animals have a Timm of ~50 min). Thus, the absolute change in phenotype with heat hardening seems like it might actually be pretty similar between small and large animals. This suggests to me that the actual rate of change is not that different, even using the definition of rate adopted by the authors. Since this was a bit of back of the envelope estimate, I can't say how true this conclusion may be, but I think it is critically important that the authors carefully examine their raw data to see whether this is actually the case. In addition, if we ask the question of the extent to which the heat-hardened animals achieve the fully acclimated state, the small animals actually get much closer to the final state than do the larger animals. Taking this perspective could completely change the conclusions drawn.

See response to previous comments 4 & 5.

Specific comments

7. Line 68-69: It is not entirely clear what is meant by developmental variation in the following sentence "because they develop directly after hatching, meaning that differences in size are not confounded with developmental variation". I assume that you mean to imply differences among instars or other clearly morphologically different developmental stages, but I could easily imagine that there could be substantial metabolic or biochemical differences over time during development in otherwise morphologically similar individuals, so I am not sure why having a direct developer necessarily controls for this issue. Indeed, at line 71-72 it is stated that mass-specific metabolic rate and heat tolerance vary with body size, which suggests that there is a direct effect of developmental stage on the physiology of these organisms. So I think this statement needs to be clarified or expanded upon a bit. Ultimately, direct development does not allow you to disentangle the effects of size from the effects of age, unless you have different clones that have different growth rates.

We agree with Referee 1 and have clarified our meaning in this statement, chiefly that size in Daphnia is not confounded with major transitions in morphology/life stage, as it is in many invertebrates (lines 68 – 70).

8. Line 73: The cited paper is under review and not available to the referee. What is important here is the rate of acclimation relative to the rate of change in size during growth. This is impossible for a referee to assess without access to these two pieces of information.

The paper cited here is now available online. We have also expanded this sentence so that the reader has more of an idea as to how rapidly heat tolerance can be adjusted in Daphnia (lines 72 – 75). Information (and citation) pertaining to the rate of growth for the genotype used in this study is present in the methods (lines 169 – 172).

9. Line 95: The length of this fixed period seems like it would be critical, as different results might be obtained with different lengths of time (particularly if rates of acclimation are non linear). We are only told at Line 134 – 9-10 hours was chosen. How was this length of time determined? In particular, cold acclimation and warm acclimation are known to proceed at different rates in a variety of taxa. Why was the same amount of time chosen for both the heat hardening and the cold hardening experiments?

The length of the hardening period was chosen based on our own data (referred to in response to previous comment). The same duration of the heat hardening and cold hardening was chosen to minimize the possibility that body sizes might change more in one treatment group rendering results from the two treatments less comparable. See lines 107 – 109 & 146 – 147.

10. Line 151: L space missing between the word “stereomicroscope” and the word “as”

Typo has been corrected.

11. Line 198: What is known about the effect of photoperiod on heat tolerance in Daphnia? Does the fact that these animals were held at 24h light reduce any photoperiod effects?

We are unaware of any literature in Daphnia describing the relationship between photoperiod and heat tolerance. Nevertheless, we don't see this as an issue in the current study because all individuals experienced the same photoperiod prior to measurement and the photoperiod was designed to mimic that experienced by the genotype during summer (genotype was sourced from a pond located within the arctic circle, this information is on lines 123 - 125)

12. Line 208: It is not clear to me that log transformation is necessary/appropriate. This choice needs to be justified (see my general comments) Also, I think it is important to be clear that you actually used natural log transformation, not log base 10. Again, this choice should be justified. There is substantial potential for deceiving yourself with respect to the patterns when looking at the log transformed data (particularly as presented in Figure 3 because high values are “squished” together on the y-axis, while low values are spread out, which can be a bit deceptive visually.

We have clarified that we used natural logarithm transformation. See Methods & Figure 3. See response to comment 5 for justification for choosing logarithmic transformation of the data.

13. Line 224: I am not clear that Table 1 is necessary

Table 1 has been deleted.

14. Line 256-257: I disagree with the interpretation of the results presented here: “it was relatively large individuals who were able to increase Timm more than small individuals, indicating a faster rate of acclimation in heat”. I do not think the data necessarily support this interpretation of the data. See my general comments.

Text in the results has been amended to reflect new results arising from the definition of acclimation rate proposed by Referee 1 (see lines 270 – 273).

15. Line 269: The axes on Figure 3 should be clearly labelled to indicate that this is a natural logarithm (either by using log base e notation or ln).

We have adjusted the labelling of the axes in Figure 3 as suggested.

16. Line 279: I think the untransformed data should be plotted in the supplemental material, and the data repository in FigShare should probably be referenced in the legend. I would also like to see an additional data supplement that includes the calculated Timm, so that the reader does not necessarily need to start with the raw data, but can simply examine the Timm data that you used to generate this figure and the untransformed data.

A plot of the untransformed data is now contained in the supplement (Fig S2). In the legend of Figure 3, we have included a private link to the Timm data that is now available in the FigShare project for this manuscript. Should the manuscript ultimately be accepted for publication all data and code contained in this FigShare project will be migrated to Dryad as provided by Royal Society publishing

17. Line 308: “fine-tune” should be “fine-tunes”

Typo has been corrected.

Referee: 2

This is a potentially publishable manuscript that shows that acclimation rates and capacities vary within a species of clonal zooplankton with age/development/body size. The manuscript is well-written and the research appears to be well done. Furthermore the question is important and interesting.

1. However, the problem with the manuscript is that the authors completely confound age with body size and thus cannot conclude that body size is the driver. The authors finally acknowledge this in the Discussion but the entire framing of the paper and the abstract and title are based strictly on body size with no mention of development/age. Ironically, most of the Discussion focuses on development/age rather than body size, despite development/age not being mentioned before the Discussion. If the capacity for phenotypic plasticity develops with age, much like other physiological capacities, such as immunity, then it is not surprising that younger, smaller individuals acclimate less and at a slower rate. The authors also bring up other development/age hypothesis in the Discussion. In the absence of evidence against this being a developmental response, I simply cannot condone such a strong emphasis being placed on body size throughout the manuscript because that data do not support this emphasis. Thus, the Title, Abstract and Introduction would need to be revised so they address these three hypotheses (age, development, body size) in a more balanced manner.

We thank Referee 2 for pointing out this bias. We have amended text in the Title, Abstract and Introduction accordingly.

2. One possible way around age and body size being confounded would be to attempt to decouple them experimentally, but this can be difficult and might not even be possible. The authors could

attempt to do this by feeding clones that hatch at the same time different amounts of food, but I suspect that the ones that get more food will develop more quickly despite being of the same age as the other group. Nevertheless, this could help address this issue.

We wholeheartedly agree with the principle here, but in our opinion decoupling age and body size experimentally, e.g. by dietary manipulation, would run the risk of introducing further physiological variation to the measurements presented here, for example via 'programming' of metabolism as a response to food deprivation/over-abundance in early life.

3. Title: The title is a bit misleading. It gives the impression of generality, but the study is only on intraspecific variation within a single species. The title should be revised to make clear the narrow scope of the study and to reduce the emphasis on body size only.

We have adjusted the title as suggested.

4. L 337-341: Yes, but a crucial distinction between the inter- and intraspecific comparisons is that the former does not confound age/development with body size and the latter does. So, comparing the two is like comparing apples with oranges. Making this clear here is very important.

We have deleted text relating to this comment and comments 5 and 6 below. Text from the Introduction relating to theme addressed in the passage of text referred to here and in comments 5 and 6 has also been removed.

5. L 341-343: You cannot conclude that mass-specific metabolic rate is not a driver of the interspecific comparisons if all or most of the intraspecific comparisons are confounded. Please revise this sentence accordingly.

See response to comment 4.

6. L 345-349: Again, you have no evidence to support the notion "that different mechanisms may be driving the relationships between acclimation rate and body size observed at each level of organization." Almost all of the intraspecific studies confound age/development with body size. So, differences between inter- and intraspecific studies are just as likely to be due to this confounder as true underlying differences in mechanisms. Please address.

See response to comment 4.

7. L 372-376: I appreciate this ending to the manuscript.

Appendix B

Dear Professor Hutchinson,

We wish to thank both Referees for reading our resubmission and in particular Referee 2 for their additional insightful comments. We have revised the manuscript accordingly and address each comment (numbered) below. In each case, we list the original comment and follow it with our response in italics. A copy of the manuscript with our revisions made since the previous version marked as 'tracked changes' follows our responses to the referee comments. Where line numbers are referenced in our responses, they refer to line numbers in the main version of the revised manuscript (not the tracked changes version).

Associate Editor Board Member

Comments to Author:

We have received feedback on your resubmission to Proceedings B. While one reviewer had no further comments for you to address, the other reviewer provided a number of points that were not fully addressed in this version. I would appreciate if you could please deal with these in a revised version, paying particular attention to the concerns regarding the interpretation of the data as being driven by large versus small individuals compared to differences in life history stages.

Reviewer(s)' Comments to Author:

Referee: 2

Comments to the Author(s).

The authors have adequately addressed my concerns. The manuscript is well-written and the research appears to be well done. Furthermore the question is important and interesting.

Referee: 1

Comments to the Author(s).

General comments:

1. The paper is improved relative to the previous version, and I greatly appreciated the inclusion of the non-transformed data in the supplement, although I would have appreciated a bit more analysis of the data in this form (e.g. a non linear fit to the data). The paper is generally very well written. The question addressed is interesting, and the work has been well performed, although the fundamental confound between size and age remains.

We have applied non-linear regression to the data contained in the supplement. See revised Supplementary Material.

2. However, I still struggle with some of the interpretation of the data. Although I agree with the vast majority of the authors' interpretation of their data, I am not sure that I agree with the section of the discussion around lines 340-347. In this section of the discussion, the authors make the argument that younger/smaller animals may be viewed as generalists with the ability to rapidly respond to environmental change via fast acclimation, while larger/older individuals may be viewed as specialists that may be more susceptible to rapid environmental change. But to me this argument seems to entirely ignore the differences in acclimation capacity between the two life stages (discussed at line 313-316), since younger/smaller individuals show limited acclimation capacity and high overall hardiness and larger/older individuals show substantial

acclimation capacity, but lower overall hardiness. I suppose this issue comes down to exactly what you mean by generalist vs specialist. Is a non-plastic but highly tolerant individual a generalist because it can cope with most of the environments it is likely to encounter? Or is a plastic but less tolerant individual a generalist because it can adopt multiple phenotypes? I think this actually highlights a logical flaw in setting this up as a binary choice. In fact, in much of the plasticity literature, I think that at least three alternative strategies are usually envisaged (generalist, specialist, and plastic). See for example, Phenotypic Plasticity: Functional and Conceptual Approaches, DeWhitt and Scheiner. While this seems to be a bit of a semantic issue, I think it actually is rather fundamental in terms of the way the data will be interpreted. Are young/small individuals likely to be able to cope with environmental change because they have the capacity to rapidly (but very marginally) change their phenotype, or simply because they are very hardy to start with? This makes a big difference in what you take away from the study.

We agree with Referee 1. We have deleted reference to the terms 'generalist/specialist' and re-structured a large part of the Discussion in line with this comment. See lines 280 – 300.

3. I think this also impinges on the overall framing of the study in the introduction, which still has at least some focus on the previously observed inter-specific pattern of differences in rates of acclimation between large and small individuals (although this is somewhat reduced relative to the previous version). Unfortunately, I think this experiment has very little to say with respect to this question, as the patterns in *Daphnia* may simply reflect different strategies adopted at different life history stages, rather than by individuals of different sizes. Indeed, the observed patterns in the data seem more consistent with a life-history based hypothesis (i.e. producing young that are ready for anything, but having older individuals that can shape their phenotype through plasticity in response to the prevailing environment). Overall, I think the paper would be better framed as addressing whether acclimation capacity and/or rate varies across development than whether acclimation capacity and/or rate varies with body size, given the fundamental confound between these two issues with this experimental design.

We have re-phrased the Introduction (and part of the Abstract) to shift the focus away from the relationship between acclimation and body size toward the more general relationship between acclimation and development. Reference to macrophysiological patterns in acclimation has been deleted in the Abstract and Introduction. See Abstract and lines 34 – 45.

Minor specific comments

4. Line 22-23: I prefer to see a somewhat more quantitative and detailed description of the results, rather than just the conclusions from them, particularly in a case where I think the interpretation of the results is somewhat open to question.

We have expanded the description of our main results in the abstract, so that they are more open to interpretation by the reader.

5. Line 38: I am not sure I agree with the statement “measurements of the rate at which [acclimation] progresses within-species are surprisingly rare”. First, any single study of the rate of acclimation is necessarily a within-species examination, so I am not sure what this sentence is trying to say. Second, there are actually a rather large number of studies of the rate of acclimation of various traits (especially in fish). Perhaps this sentence was intended to read “measurements of variation in the rate at which acclimation progresses within species are surprisingly rare”. This, I think, is a fairly true statement.

We have clarified this statement as suggested. See lines 40 – 41.

6. Line 107-109: I am not sure the logic of choosing 40% of the time for “full acclimation” is as solid as this statement implies. It seems to me to be very dependent on the reliability of your estimate of the time to full acclimation. For example, if you rely on data that were taken at only a single time point (after some longish period of acclimation), you have no idea how long prior to that time point “full acclimation” was reached. As a result, you may be back calculating your 40% time point from the wrong spot. I think I must be missing something in the logical chain. Perhaps expand the justification of this choice a bit more.

We have clarified the logic behind choosing a hardening period of ca. 10 hours by expanding/restructuring the text referred to by Referee 1 here. See lines 92 – 98.

7. Line 340-346: As mentioned in the general comments, I struggle with the logic of this section of the discussion.

See response to Comment 2 by Referee 1